# Nuclear S6K1 Enhances Oncogenic Wnt Signaling by Inducing Wnt/β-Catenin Transcriptional Complex Formation

**DOI:** 10.3390/ijms232416143

**Published:** 2022-12-18

**Authors:** Min Gyu Lee, Hwamok Oh, Jong Woo Park, Jueng Soo You, Jeung-Whan Han

**Affiliations:** 1School of Pharmacy, Sungkyunkwan University, Suwon 16419, Republic of Korea; 2Department of Biochemistry, School of Medicine, Konkuk University, Chungju 27478, Republic of Korea

**Keywords:** p70 S6K1, mTOR, β-catenin, Wnt signaling, Wnt-transcriptional complex

## Abstract

Ribosomal protein S6 kinase 1 (S6K1), a key downstream effector of the mammalian target of rapamycin (mTOR), regulates diverse functions, such as cell proliferation, cell growth, and protein synthesis. Because S6K1 was previously known to be localized in the cytoplasm, its function has been mainly studied in the cytoplasm. However, the nuclear localization and function of S6K1 have recently been elucidated and other nuclear functions are expected to exist but remain elusive. Here, we show a novel nuclear role of S6K1 in regulating the expression of the Wnt target genes. Upon activation of the Wnt signaling, S6K1 translocated from the cytosol into the nucleus and subsequently bound to β-catenin and the cofactors of the Wnt/β-catenin transcriptional complex, leading to the upregulation of the Wnt target genes. The depletion or repression of S6K1 downregulated the Wnt target gene expression by inhibiting the formation of the Wnt/β-catenin transcriptional complex. The S6K1-depleted colon cancer cell lines showed lower transcription levels of the Wnt/β-catenin target genes and a decrease in the cell proliferation and invasion compared to the control cell lines. Taken together, these results indicate that nuclear S6K1 positively regulates the expression of the Wnt target genes by inducing the reciprocal interaction of the subunits of the transcriptional complex.

## 1. Introduction

The Wnt signal transduction pathway is evolutionarily highly conserved in animal species, from Drosophila to Homo sapiens. Wnt signaling plays a central role in various cellular processes, such as cell proliferation, survival, polarity, migration, and differentiation [1]. The aberrant regulation or mutation of the Wnt pathway is often linked to birth defects, cancer, and other diseases. In particular, the Wnt/β-catenin signaling pathway is a major regulator of cell proliferation. Therefore, many cancer-related studies have targeted the Wnt/β-catenin pathway. When secreted, Wnt binds to the receptor protein Frizzled (Fz) [2,3] and low-density lipoprotein-related protein 5/6 (LRP5/6), which is a co-receptor of Fz [4,5], and transduces signals to the cytoplasmic phosphoprotein Dishevelled (Dsh/Dvl). The activated Dsh/Dvl disrupts the Axin-based complex to which the ubiquitin ligase APC and GSK3 belong, resulting in β-catenin remaining undegraded by ubiquitination and translocating into the nucleus. Thereafter, translocated β-catenin interacts with transcription factor 4 (TCF4), B-cell CLL/lymphoma 9 (BCL9), p300, and Pygo2 to form a nuclear Wnt transcriptional complex [6,7], which induces the transcriptional expression of c-Myc and cyclin D1. These transcripts are closely associated with cell cycle regulation. Therefore, deregulated Wnt signaling has catastrophic consequences, eventually leading to the progression of cancer [8]. Most colorectal cancers carry a truncated version of the APC tumor suppressor gene. This renders the APC unable to bind to Axin or degrade β-catenin [9], resulting in the aberrant activation of Wnt signaling and the continuous expression of Wnt target genes.

The mammalian target of rapamycin (mTOR) is a key regulator of various intracellular and extracellular signaling processes [10,11,12]. The most significant downstream effector, ribosomal protein S6 kinase 1 (S6K1), is known to play a critical role in nucleotide and protein synthesis, which is essential for cell growth and proliferation [13,14,15]. S6K1 has been mainly studied for its cytoplasmic roles because its major isoform, p70 S6K1, was incorrectly believed to localize only in the cytosol due to the lack of a nuclear localization sequence (NLS). However, several studies have shown that S6K1 is localized to the nucleus [16,17]. Our previous studies demonstrated that nuclear S6K1 inhibits the expression of three different Wnt genes and adiponectin through the direct phosphorylation of histone H2BS36 and the subsequent H3K27 trimethylation by EZH2 recruitment [18,19].

Here, we report a novel role for nuclear S6K1 as a regulator of the Wnt/β-catenin signaling pathway. We found that activation of Wnt signaling leads to S6K1 phosphorylation at T389 and its subsequent nuclear translocation. Nuclear S6K1 bound to β-catenin and other nuclear Wnt transcription cofactors and regulated the expression of the Wnt target genes by affecting the formation of the Wnt transcriptional complex. Consistent with this finding, both the knockdown and inhibition of S6K1 in colon cancer cell lines downregulated the expression of the Wnt target genes and suppressed the proliferation, migration, and transformation of cancer cells. Collectively, the nuclear Wnt/β-catenin signaling in HEK293T cells and colon cancer cell lines is mediated by the nuclear translocation of S6K1.

## 2. Results

### 2.1. S6K1 Is Activated by Wnt Signaling Activation and Translocated into the Nucleus

To confirm the relationship between Wnt and mTOR signaling [20], we examined whether the phospho-S6K1 (Thr389) levels change following treatment with LiCl, an agonist of canonical Wnt signaling that inhibits GSK3, and the Wnt-3a ligand at specific time points. Simultaneously with the accumulation of β-catenin, S6K1 was gradually phosphorylated at Thr389 upon activation of the Wnt signaling (Figure 1a). Based on the previous research [18], we hypothesized that S6K1 might be translocated from the cytosol to the nucleus upon phosphorylation at Thr389. Through nuclear fractionation and confocal imaging, we found that S6K1 translocated into the nucleus and activated β-catenin upon LiCl treatment (Figure 1b,c). However, even when the Wnt signaling was activated, the mTOR inhibition by rapamycin treatment suppressed the nuclear translocation of S6K1 (Figure 1d). These results are consistent with previous findings that phosphorylation at Thr389 is required for the nuclear translocation of S6K1. Taken together, these findings suggest that S6K1 translocates into the nucleus when activated by the Wnt and mTOR signaling.

### 2.2. S6K1 Interacts with β-Catenin

Many studies have reported a possible association between the mTOR and Wnt signaling pathways [21]. However, direct interactions between the pathway subunits have not yet been elucidated. To verify the interaction, we performed co-immunoprecipitation (co-IP) to examine whether S6K1 interacts with β-catenin. Notably, endogenous S6K1 co-immunoprecipitated with β-catenin (Figure 2a,b). Moreover, Flag-β-catenin and Myc-S6K1 were expressed either individually or in combination in HEK293T cells. Myc-S6K1 was detected in immunoprecipitate with anti-Flag antibodies (Figure 2c). Moreover, an in vitro GST pulldown assay confirmed the specificity of the binding (Figure 2d). We further aimed to examine how the S6K1 activation affects its interaction with β-catenin. Therefore, we generated transiently transfected cells with Flag-tagged wild-type (S6K1-WT), constitutively active (S6K1-CA), and dominant-negative S6K1 (S6K1-DN) constructs. The co-IP experiments using anti-Flag antibodies showed an enhanced interaction of S6K1-CA with β-catenin compared to that of S6K1-WT and a reduced interaction of S6K1-DN (Figure 2e). Moreover, the Wnt activation significantly increased the interaction between S6K1 and β-catenin in the nuclear fraction, whereas the cytoplasmic binding between the two proteins was reduced (Figure 2f). Collectively, S6K1 directly binds to β-catenin under the influence of S6K1 activity, and its binding occurs more strongly in the nucleus depending on the Wnt activation.

### 2.3. S6K1 Regulates Wnt/β-Catenin-Mediated Transcription

The nuclear role of S6K1 in regulating transcription has been reported previously [18,19]. Because the interaction between S6K1 and β-catenin increased after the Wnt activation, we hypothesized that there might be a novel role for nuclear S6K1 in the regulation of nuclear Wnt/β-catenin signaling. To investigate whether S6K1 modulates the Wnt/β-catenin-mediated transcription, we stably knocked down S6K1 in HEK293T cells (Figure 3a) and performed a TOPFlash luciferase reporter assay, which measures the β-catenin/TCF-mediated transcription. The S6K1 deficiency significantly inhibited the β-catenin/TCF-mediated transcription, induced by LiCl, in the HEK293T cells (Figure 3b). Next, we performed a qPCR under an S6K1-deficient condition. The increased expression of the Wnt target genes, mediated by the Wnt activation, was decreased upon the S6K1 knockdown (Figure 3c). We performed the TOPFlash reporter assay after treating the cells with mTOR inhibitor rapamycin or S6K1 inhibitor PF-4708671. We found that treatment with rapamycin or PF-4708671 lowered the TOPFlash activity that was increased by the LiCl treatment (Figure 3d). Consistent with the data from the TOPFlash assay, the expression levels of the β-catenin target genes were also decreased by rapamycin and PF-4708671 treatment (Figure 3e). Likewise, the ectopically expressed S6K1-CA enhanced the TOPFlash activity, whereas S6K1-DN reduced it (Figure 3f). The mRNA expression of the Wnt/β-catenin target genes was increased by S6K1-CA and decreased by S6K1-DN, similar to the expression pattern observed in the TOPFlash assay (Figure 3g). Taken together, S6K1 regulated the Wnt/β-catenin target gene expression in an activity-dependent manner.

### 2.4. S6K1 Does Not Affect β-Catenin

Because S6K1 regulates the Wnt/β-catenin-mediated transcription, and Wnt-dependent S6K1 activation enhances its binding to β-catenin, we hypothesized that S6K1 could directly phosphorylate β-catenin. However, the in vitro kinase assay showed that S6K1 does not directly phosphorylate β-catenin, whereas it phosphorylates histone H2B as shown in our previous study. (Figure 4a) [18]. Furthermore, β-catenin accumulation, which is induced by Wnt activation, was maintained at the same level regardless of the rapamycin treatment. (Figure 4b). As indicated previously, the nuclear accumulation of S6K1 decreased in the presence of rapamycin (Figure 1d). However, the inhibition of S6K1 by rapamycin and PF-4708671 did not influence the nucleocytoplasmic localization of β-catenin (Figure 4c). The acetylation of lysine 49 of β-catenin, mediated by CBP, was not altered by the ectopic expression of S6K1-CA or DN (Figure 4d). In contrast, the β-catenin knockdown did not significantly affect the S6K1 activation by LiCl (Figure 4e).

### 2.5. S6K1 Affects the Formation of the Wnt/β-Catenin Transcriptional Complex

Nuclear β-catenin activated by Wnt signaling interacts with multiple binding partners, such as T-cell factor 4 (TCF4), E1A-binding protein p300 (p300), Pygopus family PHD finger 2 (Pygo2), and B-cell CLL/lymphoma 9 protein (BCL9), which comprise the transcriptional complex expressing Wnt target genes. Because the stability and localization of β-catenin did not change and were independent of the S6K1 activation (Figure 4b,c), we wondered how S6K1 affects the Wnt target gene expression. We investigated whether S6K1 affects the formation of the Wnt/β-catenin transcriptional complex. Interestingly, the S6K1 deficiency reduced the interaction between TCF4 and β-catenin without altering their protein levels (Figure 5a,b). We next investigated whether the interaction between other Wnt/β-catenin transcriptional complex cofactors was influenced by S6K1 inhibition. We performed a co-immunoprecipitation assay using the Pygo2 antibody and only the nuclear fraction of cell lysates. The interaction between the transcription cofactors constituting the transcriptional complex was induced by LiCl treatment but was reduced by the S6K1-specific inhibitor PF-4708671 (Figure 5c). Even when co-IP was performed with the antibody of the other transcription cofactor p300, the S6K1 inhibition suppressed the interaction of the components of the Wnt/β-catenin transcriptional complex (Figure 5d). In summary, S6K1 affected the nuclear Wnt/β-catenin transcriptional complex formation in an activity-dependent manner.

### 2.6. S6K1 Regulates Cell Proliferation and Invasiveness of Colon Cancer Cell Lines

Previous studies have reported the abnormal activation of Wnt signaling in colon cancer cells [8,22]. In addition, it has been well established that the hyperactivation of mTOR signaling, caused by the deregulation of upstream components, is a very common feature in human colon cancer [23]. In our experiments, colon cancer cell lines, such as LoVo, DLD-1, and HT-29, distinctly expressed β-catenin and p-S6K1 (T389) much more than the normal colon cell line CCD-18Co (Figure 6a). To confirm the effect of S6K1 on the Wnt signaling in colon cancer cell lines, we stably knocked down S6K1 in two cell lines, HT-29 and DLD-1 (Figure 6b). We then examined the β-catenin/TCF-mediated transcription in these two cell lines using the TOPflash reporter assay system. Consistent with the results from the HEK293T cells (Figure 4b), the downregulation of S6K1 reduced the luciferase activity in both these colon cancer cell lines (Figure 6c). In addition, it was confirmed that the expression of the Wnt target genes was reduced by decreased Wnt signaling activity (Figure 6d). Moreover, we assessed the proliferation rate in a time-dependent manner and found that the loss of S6K1 delayed the proliferation in both cell lines (Figure 6e). We performed wound-healing assays, and the results showed that the S6K1 deficiency decreased the migratory ability of these two colon cancer cell lines (Figure 6f). These results indicated that the S6K1 activation by Wnt signaling is important for mediating the transcriptional regulation of colorectal cancer cell proliferation via Wnt/β-catenin signaling.

## 3. Discussion

In this study, we found that S6K1 is translocated into the nucleus upon Wnt signaling activation and directly binds to β-catenin. Moreover, S6K1 affects the interaction between β-catenin and TCF4, as well as the binding between other Wnt/β-catenin transcriptional cofactors, such as β-catenin, Pygo2, BCL9, and p300, thereby mediating the regulation of the Wnt/β-catenin target gene expression (Figure 7). Although the mechanism by which nuclear S6K1 affects those remains elusive, it seems clear that S6K1 activity is required for the regulation of the Wnt transcriptional complex formation. It is known that S6K1 phosphorylates most substrates at the R/K-X-R-X-X-S/T (where X indicates any amino acid) motif. Interestingly, the Wnt transcription cofactors Pygo2, BCL9/9L, and p300 have S6K1 consensus phosphorylation motifs, whereas it is absent in β-catenin. For instance, a previous study about Pygo2 has reported that phosphorylation at serine 48 (p-S48) reduces the polyubiquitylation of Pygo2 and increases its stability [24]. P300 is also known to be phosphorylated by Akt at serine 1834, which is required for its histone acetyltransferase and transcriptional activity [25]. Furthermore, S6K1 and Akt are known to share the same substrates and phosphorylation sites, such as serine 9 in GSK-3 and serine 136 in BAD. Therefore, it is possible that S6K1 phosphorylates the Wnt transcription cofactors, which are substrates of Akt as well.

Here, we found that long-term treatment with PF-4708671, the specific inhibitor of S6K1, decreased the protein levels of Pygo2. If S6K1 phosphorylates Pygo2 directly, it can be hypothesized that the knockdown or inhibition of S6K1 leads to the reduced phosphorylation of Pygo2 and induces its degradation through increased ubiquitination. Consequently, this downregulation of Pygo2 can suppress the formation of the transcriptional complex that constitutes Pygo2 as a cofactor, thereby reducing the expression of Wnt target genes. To substantiate this hypothesis, further experiments are required to demonstrate that S6K1 can directly phosphorylate and regulate Pygo2.

The Wnt/β-catenin signaling pathway regulates many cellular processes, such as cell proliferation, differentiation, and migration. If signal transduction is dysregulated and homeostasis is not maintained, tumorigenesis can occur. Therefore, the Wnt/β-catenin signaling pathway has been targeted for cancer treatment [26,27]. The goal of these cancer therapeutics is to suppress the excessive expression of cancer-related Wnt target genes.

We showed that S6K1 translocates from the cytosol into the nucleus after Wnt signaling activation and directly interacts with β-catenin and several Wnt transcriptional cofactors to transcriptionally regulate the Wnt target genes in a kinase activity-dependent manner. Our results revealed that the deregulation of S6K1, which leads to the overactivation of the Wnt/β-catenin signaling, is closely linked to colorectal cancer cell proliferation and migration. It has been well established that the aberrant expression of Wnt target genes induces cancer progression. Therefore, elucidating the detailed mechanism of S6K1 deregulation might be beneficial in developing S6K1-based therapeutic approaches for colon cancer.

## 4. Materials and Methods

### 4.1. Antibodies

Anti-p70 S6K1 (Santa Cruz Biotechnology, mouse SC-8418, rabbit SC-230, Dallas, TX, USA and Cell Signaling Technology, rabbit 9202, Danvers, MA, USA), anti-phospho-p70 S6K1 T389 (Cell Signaling Technology, rabbit 9205), anti-Flag (Sigma-Aldrich, F3165, Darmstadt, Germany), anti-c-Myc (Santa Cruz Biotechnology, SC-40), anti-β-catenin (Santa Cruz Biotechnology, mouse SC-7963 and rabbit SC-7199, and BD Transduction Laboratories, 610153, Franklin Lakes, NJ, USA), anti-acetyl-β-catenin K49 (Cell Signaling Technology, rabbit 9030, 9534), anti-TCF4 (Santa Cruz Biotechnology, mouse SC-166699, rabbit SC-13027), anti-BCL9 (Cell Signaling Technology, rabbit 15096), anti-Pygopus2 (Santa Cruz Biotechnology, mouse SC-390506), anti-p300 (Santa Cruz Biotechnology, mouse SC-48343 and rabbit SC-384), anti-cyclin D1 (Santa Cruz Biotechnology, SC-8396), anti-S6 (Cell Signaling Technology, rabbit 2217), anti-histone H3 (Santa Cruz Biotechnology, rabbit SC-10809), anti-lamin A/C (Cell Signaling Technology, rabbit 2032), anti-α-tubulin (Santa Cruz Biotechnology, SC-32293), anti-actin (Merck Millipore, Mab-1501, Temecula, CA, USA), and anti-IgG (Santa Cruz Biotechnology, mouse SC-2025 and rabbit SC-2027) were used.

### 4.2. Plasmids and Reagents

The following reagents and plasmids were used: pRK5 Myc-tagged human S6K1 wild-type (WT), constitutively active (CA), and dominant-negative (DN). pcDNA Flag-tagged human β-catenin. pGEX-4T-1-based bacterial-expressed GST-tagged empty vector and human β-catenin. Rapamycin, the mTOR specific inhibitor (Calbiochem, 553210, Darmstadt, Germany); PF-4708671, the S6K1 specific inhibitor (Tocris, 4032, Bristol, UK); Wnt-3a, Wnt activator that binds to Frizzled (R&D Systems, 5036-WN, Minneapolis, MN, USA); lithium chloride (LiCl), the GSK-3 inhibitor that acts as a Wnt activator (Sigma Aldrich, L9650, St. Louis, MO, USA); and active p70 S6K1 protein for in vitro kinase assay (SignalChem, R21-10H, Richmond, BC, Canada).

### 4.3. Cell Culture and Drug Treatment

HEK293T cells were grown in Dulbecco’s modified Eagle medium (DMEM, Welgene, Gyeongsan, Republic of Korea). HT29 cells were grown in McCoy’s 5A medium (Welgene). CCD-18Co cells were grown in minimum essential medium (MEM, Welgene). DLD-1 cells were grown in Roswell Park Memorial Institute (RPMI) 1640 medium (Welgene). LoVo cells were grown in Ham’s F-12K (Kaighn’s) medium (Ham’s F-12K, Welgene). All cells were grown in media supplemented with 10% fetal bovine serum (FBS, Welgene) and 1% penicillin/streptomycin (P/S, Welgene). Cells were maintained in a humidified atmosphere of 5% CO_2_ at 37 °C.

Cells were treated with rapamycin (25 nM) or PF-4708671 (10 μM) to inhibit mTOR or S6K1 activities, respectively, or the same amount of DMSO as a control. Cells were treated with Wnt-3a (300 ng/mL) or LiCl (25 mM) for the indicated time points.

### 4.4. Protein Extraction and Immunoblotting

For immunoblotting, whole-cell proteins were extracted using PRO-PREP protein extraction solution (iNtRON Biotechnology, Seongnam, Republic of Korea). The cell lysates were centrifuged for 20 min at 16,000× *g* at 4 °C. The supernatant was then transferred into a new tube. For nuclear fractionation, cells were lysed in harvest buffer (10 mM HEPES (pH 7.9), 50 mM NaCl, 0.5 M sucrose, 0.1 mM EDTA, 0.5% Triton X-100, and freshly added DTT, PMSF, and protease inhibitors), incubated on ice for 5 min, and centrifuged at 120× *g* for 10 min at 4 °C. The supernatant (cytosolic fraction) was transferred into a new tube. The nuclear pellet was washed twice with 500 μL of buffer A (10 mM HEPES (pH 7.9), 10 mM KCl, 0.1 mM EDTA, and 0.1 mM EGTA) and centrifuged at 120× *g* for 10 min at 4 °C. The supernatant was discarded, and the pellet (nuclear fraction) was resuspended in PRO-PREP for lysis.

Equal amount of each protein sample was used. Proteins were separated by sodium dodecyl sulfate-polyacrylamide gel electrophoresis. Then, they were transferred onto polyvinylidene difluoride membranes (Merck Millipore, Temecula, CA, USA) using the wet transfer method. The membranes were incubated with the indicated primary antibodies overnight at 4 °C. Subsequently, the membranes were incubated with horseradish peroxidase (HRP)-conjugated secondary antibodies (Merck Millipore) for 1 h at room temperature. The HRP signals were detected using AbSignal (Abclon, Seoul, Republic of Korea).

### 4.5. Immunofluorescence

Cells were cultured in 12-well plate on gelatin-coated cover glass. At 0, 4, and 8 h after LiCl treatment, samples were fixed with 4% paraformaldehyde for 15 min, and rehydrated in PBS for 10 min. Cell permeabilization and blocking process was performed with PBS-BT buffer (3% BSA, 0.05% NaN_3_, and 0.1% Triton X-100 in PBS) at room temperature for 30 min. Primary and secondary antibodies were diluted in PBS-BT at an appropriate ratio and incubated at room temperature for 1 h at each step in a humidified chamber. Staining was performed sequentially from the primary to the secondary. Between each step, samples were washed twice with PBS-BT buffer.

### 4.6. Immunoprecipitation

Cells were harvested and lysed using PRO-PREP (iNtRON Biotechnology, Seongnam, Republic of Korea) or IP lysis buffer (40 mM HEPES (pH 7.4), 120 mM NaCl, 1 mM EDTA, 50 mM NaF, 1.5 mM Na_3_VO_4_, 10 mM β-glycerophosphate, 0.3% CHAPSO, and 0.1% protease inhibitor cocktail). The indicated specific antibodies and the agarose beads for the type of antibodies were added to equal amount of each protein sample and incubated at 4 °C overnight. The beads were then centrifuged briefly and washed three times with IP wash buffer (IP lysis buffer without CHAPSO). Immunoprecipitated proteins were eluted with 2× sample buffer (Bio-Rad, Hercules, CA, USA) by boiling at 95 °C for 5 min. The eluted protein samples were analyzed by immunoblotting.

### 4.7. RNA Extraction and Quantitative Real-Time PCR (qRT-PCR)

Whole-cell RNA was purified using Easy-Blue reagent (iNtRON Biotechnology, Seongnam, Republic of Korea) according to the manufacturer’s instructions. For cDNA synthesis, 1 μg of purified RNA and Maxime RT premix (iNtRON Biotechnology) were used. qRT-PCR was performed on the CFX96 Real-Time PCR System (Bio-Rad, Hercules, CA, USA) using KAPA SYBR^®^ FAST qPCR Master Mix (Kapa Biosystems, Wilmington, MA, USA). The relative expression levels of the target genes were normalized to the expression levels of GAPDH and β-actin. The primer pairs used for qRT-PCR are listed in Table 1.

### 4.8. TOPFlash Luciferase Assay

Cells were cultured at an equal density of 5 × 10^4^ cells/well in 12-well plates. The cells were transfected with TOPFlash reporter (0.5 μg) using Lipofectamine 2000 (Thermo Fisher Scientific, 11668-019, Waltham, MA, USA). Cells pretreated with specific inhibitors or overexpressing Myc-S6K1 (WT, CA, and DN) or stably expressing S6K1 shRNA were treated with 20 mM LiCl for the indicated time points. Reporter activity was measured using the Dual-Luciferase Reporter Assay System (Promega, Madison, WI, USA) according to the manufacturer’s instructions.

### 4.9. Cell Proliferation and Migration Assay

Cells were equally seeded on each plate, and samples were obtained by trypsinizing one plate from each group daily for counting the number of cells. Each sample was counted using Luna-II™ automated cell counter according to manufacturer’s instructions.

HT-29 and DLD-1 cells were seeded in 6-well plates at 80% confluency. After 24 h, wounds were created by scraping with pipette tip, and the cells separated from the plate were removed by washing twice with medium. Migration images were taken at 0, 1, and 2 days after wound creation. 

### 4.10. Statistical Analysis

Statistical analysis was performed using GraphPad Prism (version 7.0) (Graphpad Software Inc., San Diego, CA, USA). Statistical significance was calculated by performing Student’s *t*-test and results with * *p* < 0.05, ** *p* < 0.01, *** *p* < 0.001 were considered significant (n.s.: no significance). 

## Figures and Tables

**Figure 1 ijms-23-16143-f001:**
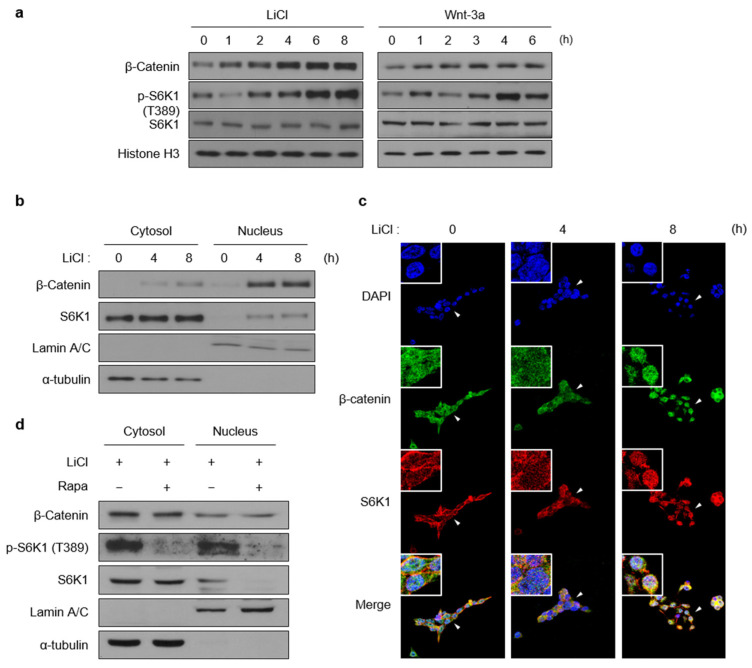
Wnt signaling pathway activates S6K1 and induces its translocation into the nucleus. (**a**) Wnt-3a and LiCl induce S6K1 phosphorylation. Cells (293T) were treated with Wnt-3a (300 ng/mL) and LiCl (25 mM) for the indicated time points. Immunoblotting was performed using specific antibodies as indicated. (**b**,**c**) LiCl promotes S6K1 nuclear translocation. Immunoblotting was performed using cytoplasmic and nuclear extracts from 293T cells treated with LiCl for the indicated time points (**b**). Localization of S6K1 and β-catenin was detected by immunostaining 293T cells treated with LiCl for the indicated time points (**c**). (**d**) Rapamycin inhibits Wnt-induced S6K1 nuclear translocation. Immunoblotting was performed using cytoplasmic and nuclear extracts from 293T cells treated with LiCl in the absence or presence of rapamycin (25 nM). Cells were pretreated with specific inhibitors for 1 h before LiCl treatment.

**Figure 2 ijms-23-16143-f002:**
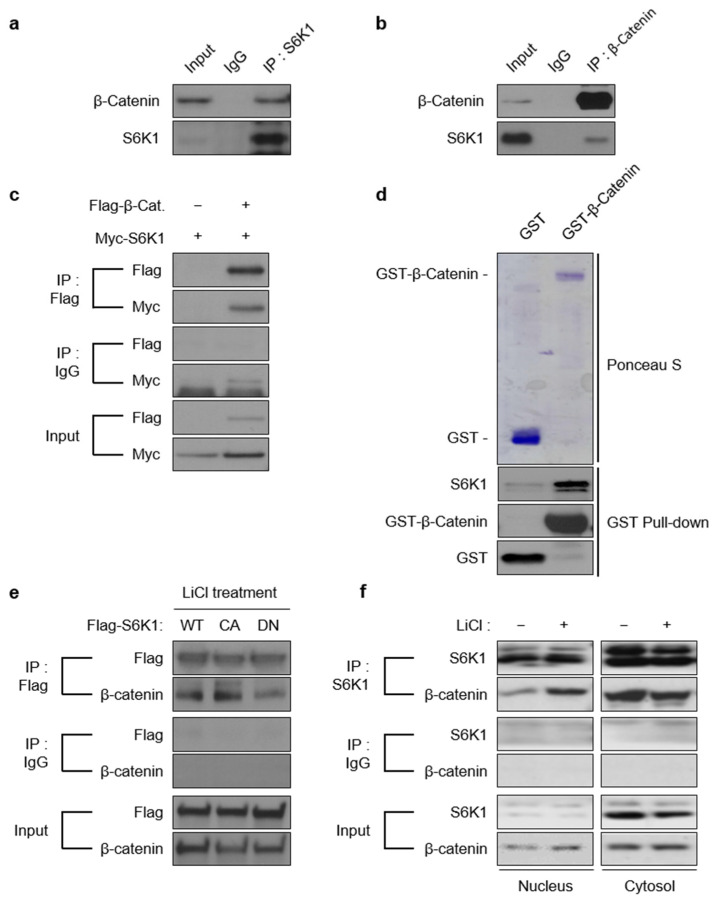
S6K1 binds directly to β-catenin. (**a**,**b**) Endogenous S6K1 binds to β-catenin. Co-immunoprecipitation assays were performed in DLD-1 cells using S6K1 (**a**) and β-catenin (**b**) antibodies. Each binding partner was detected by immunoblot performed with the indicated antibodies. (**c**) Exogenous S6K1 binds to β-catenin. Co-immunoprecipitation assay was performed using Flag-antibody and 293T cells that were transiently transfected with exogenous Flag-β-catenin and Myc-S6K1. Immunoblot was performed with anti-Flag and anti-Myc antibodies. (**d**) S6K1 binds to β-catenin in vitro. In vitro GST pulldown assay was performed using recombinant GST or recombinant GST-β-catenin with 293T cell extracts. Presence of the GST and GST-fusion proteins were confirmed by Ponceau staining (**top**). Eluted S6K1 and GSTs were detected by immunoblot performed with anti-S6K1 and anti-GST antibodies (**bottom**). (**e**) S6K1 binds to β-catenin in a kinase activity-dependent manner. Co-immunoprecipitation assay was performed using Flag-antibody and 293T cells stably expressing Flag-tagged WT, CA, and DN forms of S6K1 treated with LiCl (25 mM) for 8 h. (**f**) The interaction between nuclear S6K1 and β-catenin was increased in a Wnt activation-dependent manner. Co-immunoprecipitation assay was performed using S6K1 antibody and cytoplasmic and nuclear extracts from 293T cells treated or untreated with LiCl (25 mM, 8 h).

**Figure 3 ijms-23-16143-f003:**
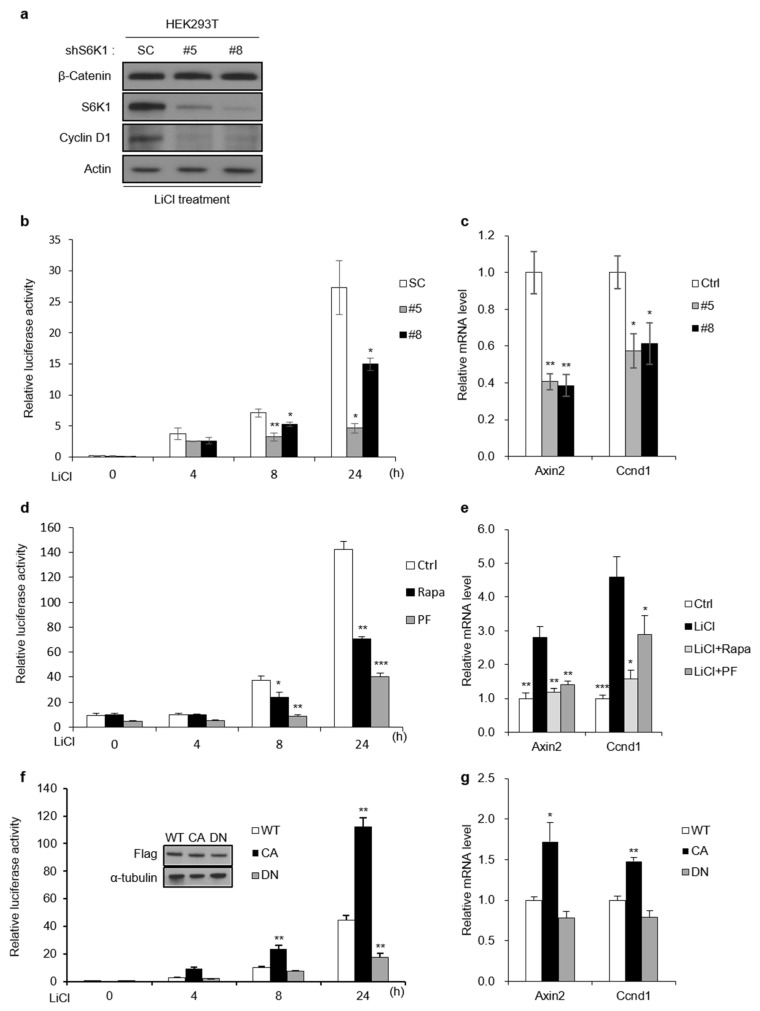
S6K1 regulates Wnt/β-catenin-mediated transcription. (**a**) Reduced S6K1 affects cyclin D1 protein expression. Immunoblotting was performed using 293T cells with stable S6K1 knockdown (293T-S6K1 KD) treated with LiCl (25 mM) for 8 h. (**b**) Reduced S6K1 suppresses TOPFlash luciferase reporter activity. The TOPFlash luciferase reporter assay was performed using 293T cells treated with LiCl for the indicated time points after transient transfection with S6K1 siRNA (48 h). (**c**) Reduced S6K1 suppresses β-catenin transcriptional activity. qRT-PCR was performed using 293T cells treated with LiCl for 8 h after transient transfection with S6K1 siRNA (48 h). (**d**) S6K1 inhibition suppresses TOPFlash luciferase reporter activity. The TOPFlash luciferase reporter assay was performed using 293T cells treated with LiCl for the indicated time points after pretreatment for 1 h with rapamycin (25 nM) and PF-4708671 (10 µM). (**e**) S6K1 inhibition suppresses β-catenin transcriptional activity. qRT-PCR was performed using 293T cells treated with LiCl for 8 h after pretreatment for 1 h with rapamycin (25 nM) and PF-4708671 (10 µM). (**f**) TOPFlash luciferase reporter activity is regulated by S6K1 in a kinase activity-dependent manner. The TOPFlash luciferase reporter assay was performed using 293T cells stably expressing Flag-tagged WT, CA, and DN forms of S6K1 treated with LiCl (25 mM) for the indicated time points. (**g**) Kinase activity of S6K1 affects β-catenin transcriptional activity. qRT-PCR was performed using 293T cells stably expressing Flag-tagged WT, CA, and DN forms of S6K1 treated with LiCl (25 mM) for 8 h. Data are presented as the mean ± SEM for *n* = 3. * *p* < 0.05; ** *p* < 0.01; *** *p* < 0.001.

**Figure 4 ijms-23-16143-f004:**
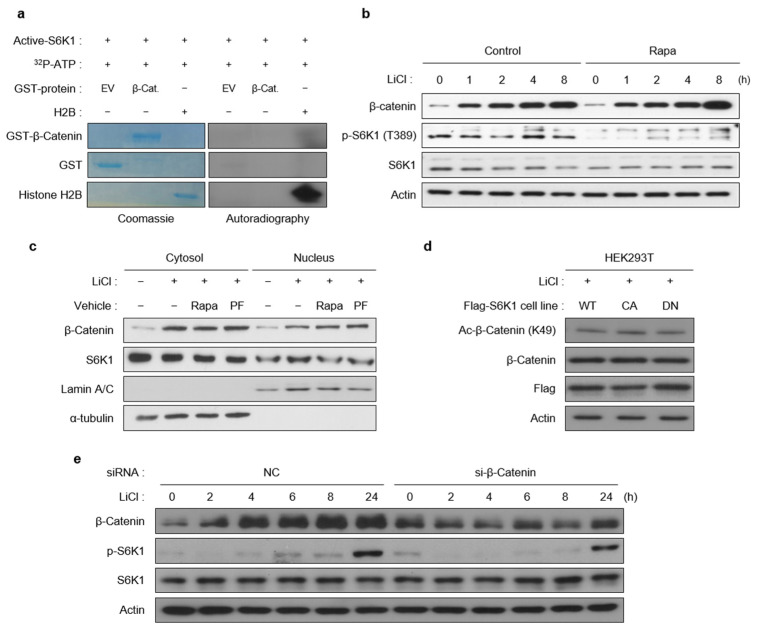
S6K1 does not affect β-catenin. (**a**) β-catenin is not a kinase substrate of S6K1. In vitro kinase assays were performed using GST, GST-β-catenin, and recombinant histone H2B as substrates and recombinant active-S6K1 as the kinase in the presence of 32P-ATP. The presence of substrates was confirmed by Coomassie blue staining (**left**), and 32P-mediated phosphorylation was detected by photosensitization with X-ray film (**right**). (**b**) Inhibition of S6K1 did not influence LiCl-induced β-catenin accumulation. Immunoblotting was performed using 293T cells pretreated with rapamycin (25 nM, 1 h) followed by LiCl (25 mM) for the indicated time points. (**c**) S6K1 inhibition did not affect the nucleocytoplasmic distribution of β-catenin. Immunoblotting was performed using cytoplasmic and nuclear extracts from 293T cells pretreated with or without rapamycin (25 nM) and PF-4708671 (10 µM) for 1 h and then treated with LiCl for 8 h.) (**d**) S6K1 does not affect the acetylation of K49 residue of β-catenin in a kinase activity-dependent manner. Immunoblotting was performed using 293T cells stably expressing Flag-tagged WT, CA, and DN forms of S6K1 treated with LiCl (25 mM) for 8 h. (**e**) β-catenin knockdown did not influence LiCl-induced phosphorylation of S6K1. Immunoblotting was performed using 293T cells treated with LiCl for the indicated time points after transient transfection with β-catenin siRNA (48 h).

**Figure 5 ijms-23-16143-f005:**
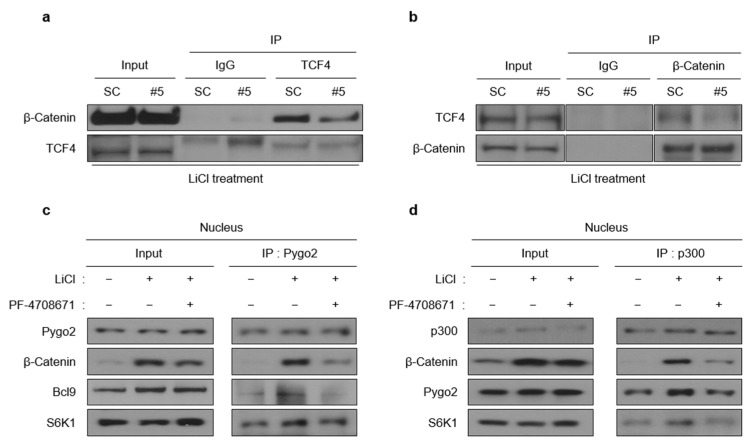
S6K1 affects the formation of the Wnt/β-catenin transcriptional complex. (**a**,**b**) S6K1 knockdown reduces the interaction between β-catenin and TCF4. Co-immunoprecipitation assay was performed using TCF4 (**a**) and β-catenin (**b**) antibodies and 293T cells treated with LiCl (25 mM) for 8 h after transient transfection with S6K1 siRNA (48 h). (**c**,**d**) Inhibition of S6K1 disrupts formation of Wnt/β-catenin transcriptional complex. Each co-immunoprecipitation assay was performed with nuclear extracts from 293T cells treated with DMSO or PF-4708671 (10 µM) followed by LiCl treatment for 8 h, using Pygo2 (**c**) and p300 (**d**) antibody, respectively.

**Figure 6 ijms-23-16143-f006:**
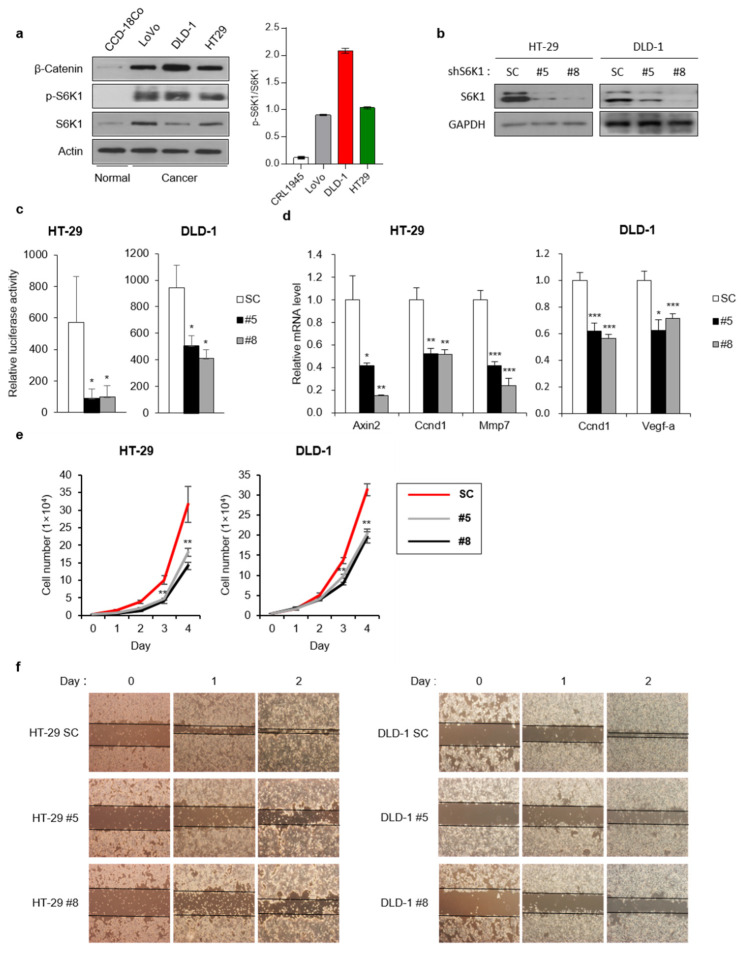
Effects of S6K1 deficiency in colon cancer cell lines. (**a**) S6K1 is aberrantly activated in colon cancer cell lines. Immunoblotting was performed using total protein extracts from CCD-18Co (normal colon cells), LoVo, DLD-1, and HT-29 (colon cancer cells) and specific antibodies as indicated. (**b**) HT-29 and DLD-1 S6K1-knockdown cell lines. Immunoblotting was performed using these S6K1-knockdown two colorectal cancer cell lines and antibodies as indicated. (**c**) Reduced S6K1 levels suppressed TOPFlash luciferase reporter activity. TOPFlash luciferase reporter assay was performed in HT-29 and DLD-1 S6K1-knockdown cell lines. (**d**) S6K1 knockdown suppresses the expression of Wnt target genes in colon cancer cells. qRT-PCR was performed in HT-29 and DLD-1 S6K1-knockdown cell lines. (**e**) Loss of S6K1 inhibits cell proliferation in HT-29 and DLD-1 colon cancer cells. Cell proliferation assay was performed in HT-29 and DLD-1 S6K1-knockdown cell lines using an automated cell counter as indicated time points. (**f**) Loss of S6K1 inhibits cell migration in HT-29 and DLD-1 colon cancer cells. Wound-healing assay was performed using HT-29 and DLD-1 S6K1-knockdown cells. Data are represented as the mean ± SEM for *n* = 3. * *p* < 0.05; ** *p* < 0.01; *** *p* < 0.001.

**Figure 7 ijms-23-16143-f007:**
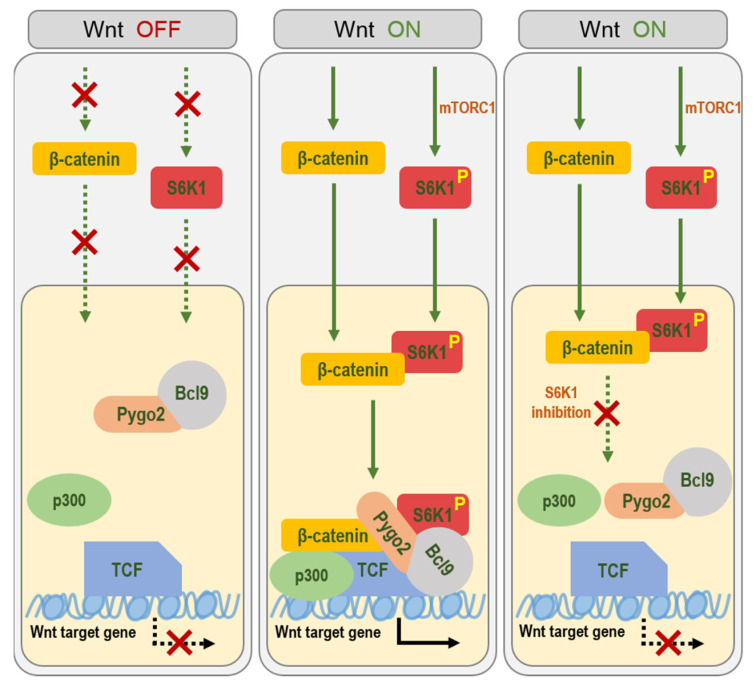
A mechanistic model.

**Table 1 ijms-23-16143-t001:** The primer pairs list for qRT-PCR.

Gene	Forward (5′ to 3′)	Reverse (3′ to 5′)
*GAPDH*	ATCATCCCTGCCTCTACTGG	GTCAAGTCCACCACTGACAC
*Axin2*	TGGACGATGTGCTCTATGCC	GGATGGTGATGGTTTGGTAG
*Ccnd1*	ATGTTCGTGGCCTCTAAGATGAA	CGGTGTAGATGCACAGCTTCTC
*Dkk1*	AAACGCTGCATGCGTCACGCTAT	AAAGCTTTCAGTGATGGTTT
*Bmp4*	CACTGGTCCCTGGGATGTTC	GATCCACAGCACTGGTCTTGACTA
*Vegf-a*	GAGTACATCTTCAAGCCATC	CATTTGTTGTGCTGTAGGAA
*β-Actin*	AAAAGCCACCCCACTTCTCT	CTCAAGTTGGGGGACAAAAA

## Data Availability

Data are contained within the article.

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
