# Peer review of "Nuclear S6K1 Enhances Oncogenic Wnt Signaling by Inducing Wnt/β-Catenin Transcriptional Complex Formation"

_ijms, 2022, doi:10.3390/ijms232416143_

Round 1
Reviewer 1 Report
The manuscript titled “Nuclear S6K1 enhances oncogenic Wnt signaling by inducing Wnt/
β-catenin transcriptional complex formation” by Lee et al identified the correlations of Wnt and S6K1. They found that activated Wnt promotes S6K1 translocated from cytosol to nucleus, thus upregulating Wnt genes. The novel mechanism of the nuclear S6K1-induced Wnt signaling revealed another important role of S6K1. This largely enriched the function of mTORC1-S6K1. However, there are some concerns as listed.
1. Wnt induces S6K1 nuclear accumulation, S6K1 promotes Wnt. Do S6K1 and Wnt signaling form feedback loop?
2. In the wnt-S6K1-Wnt axis, which is the origin step in cancer cells?
3. A mechanistic model is needed to better understand the paper.
4. In Fig1a, Fig5d, the quality of the blot is poor
5. In Fig1c, the localization of S6K1 is not clear, zoom in is needed for these pictures.
6. In Fig2, could you explain the S6 blot?
7. In Fig2b, the S6K1 signaling is too weak.
8. According to your result, Lici induced p-S6K1 has transitorily reduction. In Fig1a, ps6k1 is reduced in 1h, but in Fig4e its 2h. Could you explain the difference?
9. In Fig6e the S6K1 is not even, a better result or quantification of pS6k1/S6k1 is needed.
Author Response
Dear Reviewer,
Please kindly see the attachment.
Thank you.

Reviewer 2 Report
That is valuable and interesting work in the cell and cancer biology. Authors employed direct experiments to validate the nucleic roles of S6K1. The authors show a novel nuclear role of S6K1 in regulating the expression of Wnt target genes. That is an interesting and valuable work which is close to the clinical use and could be benefit for patients with diseases influenced by WNT related pathways, both in the diagnosis and treatment. However, several flaws should be revised before publication.
1) Authors should supplement more words about how the resource is used in their study, and the detailed parameters in the process of data analysis for independent cell lines.
2) The proteins used for WB may be not pure, then resulting in several double and dispersion stripes.
3) Several works may reproduce, e.g., the figure 5C and 5d. The concentration of beta-catetnin may be extremely higher than other treatment conditions.
Author Response
Dear reviewer,
Please kindly see the attachment.
Thank you.

Round 2
Reviewer 1 Report
The author addressed my concerns. I think it can be published.
Reviewer 2 Report
I recommend the publication of this MS